# Cold Posteriors through PAC-Bayes

**Konstantinos Pitas**
Univ. Grenoble Alpes,
Inria, CNRS,
Grenoble INP, LJK,
38000 Grenoble, France
`pitas.konstantinos@inria.fr`

**Julyan Arbel**
Univ. Grenoble Alpes,
Inria, CNRS,
Grenoble INP, LJK,
38000 Grenoble, France
`julyan.arbel@inria.fr`

## Abstract

We investigate the cold posterior effect through the lens of PAC-Bayes generalization bounds. We argue that in the non-asymptotic setting, when the number of training samples is (relatively) small, discussions of the cold posterior effect should take into account that approximate Bayesian inference does not readily provide guarantees of performance on out-of-sample data. Instead, out-of-sample error is better described through a generalization bound. In this context, we explore the connections of the ELBO objective from variational inference and the PAC-Bayes objectives. We note that, while the ELBO and PAC-Bayes objectives are similar, the latter objectives naturally contain a temperature parameter $\lambda$ which is not restricted to be $\lambda = 1$. For realistic classification tasks, in the case of Laplace approximations to the posterior, we show how this PAC-Bayesian interpretation of the temperature parameter captures important aspects of the cold posterior effect.

## 1 Introduction

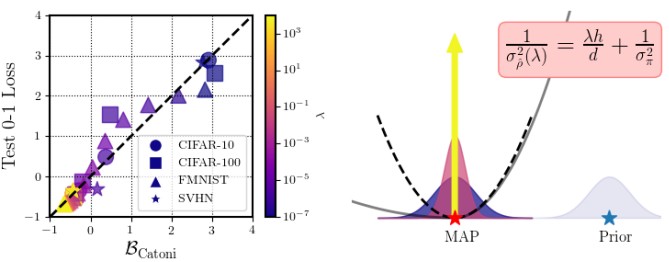

(a) Classification tasks      (b) Laplace, different temperatures $\lambda$

Figure 1: PAC-Bayes bounds correlate with the test 0-1 Loss for different values of the temperature $\lambda$ (quantities on both axes are normalized). (a) Classification tasks on CIFAR-10, CIFAR-100, and SVHN datasets ($\sigma_\pi^2 = 0.1$, ResNet22) and FMNIST dataset ($\sigma_\pi^2 = 0.1$, ConvNet). (b) Graphical representation of the Laplace approximation for different temperatures: for hot temperatures $\lambda \ll 1$, the posterior variance becomes equal to the prior variance; for $\lambda = 1$ the posterior variance is regularized according to the curvature $h$; for cold temperatures $\lambda \gg 1$, the posterior becomes a Dirac delta on the MAP estimate.

In their influential paper, Wenzel et al. [38] highlighted the observation that Bayesian neural networks typically exhibit better test time predictive performance if the posterior distribution is "sharpened" through tempering. Their work has been influential primary because it serves as a well documented

2022 Trustworthy and Socially Responsible Machine Learning (TSRML 2022) co-located with NeurIPS 2022.

example of the potential drawbacks of the Bayesian approach to deep learning. While other subfields of deep learning have seen rapid adoption, and have had impact on real world problems, Bayesian deep learning has, to date, seen relatively limited practical use [22, 27, 12, 38]. The "cold posterior effect", as the authors of Wenzel et al. [38] named their observation, highlights an essential mismatch between Bayesian theory and practice. As the number of training samples increases, Bayesian theory tells states that the posterior distribution should be concentrating more and more on the true model parameters, in a frequentist sense. At any time, the posterior is our best guess at the true model parameters, without having to resort to heuristics. Since the original paper, a number of works [35, 42, 1, 32, 15, 2, 6, 2] have attempted to explain the cold posterior effect, identify its origins, propose remedies and defend Bayesian deep learning in the process.

Here, we investigate PAC-Bayes generalization bounds [30, 9, 3, 13] as the model that governs performance on out-of-sample data. PAC-Bayes bounds describe the performance on out-of-sample data, through an application of the convex duality relation between measurable functions and probability measures. The convex duality relationship naturally gives rise to the log-Laplace transform of a special random variable [9]. Importantly the log-Laplace transform has a temperature parameter $\lambda$ which is not constrained to be $\lambda = 1$. We investigate the relationship of this temperature parameter to cold posteriors.

In summary, our contributions are the following:

- We show that PAC-Bayes bounds correlate with out-of-sample performance for different values of the temperature parameter $\lambda$.

- We find that the coldest temperature (such that the posterior is a Dirac delta centered on a MAP estimate of the weights) is empirically always optimal in terms of test accuracy.

- We derive a PAC-Bayes bound for the case of the widely used generalized Gauss–Newton Laplace approximations to the posterior. This bound might explain why it is difficult to pinpoint an exact cause for the cold-posterior effect.

We also include a detailed FAQ section in the Appendix.

## 2   Cold posterior effect: misspecified and non-asymptotic setting

We denote the learning sample $(X, Y) = \{(\boldsymbol{x}_i, y_i)\}_{i=1}^n \in (\mathcal{X} \times \mathcal{Y})^n$, that contains $n$ input-output pairs. Observations $(X, Y)$ are assumed to be sampled randomly from a distribution $\mathcal{D}$. Thus, we denote $(X, Y) \sim \mathcal{D}^n$ the i.i.d observation of $n$ elements. We consider loss functions $\ell : \mathcal{F} \times \mathcal{X} \times \mathcal{Y} \to \mathbb{R}$, where $\mathcal{F}$ is a set of predictors $f : \mathcal{X} \to \mathcal{Y}$. We also denote the risk $\mathcal{L}_{\mathcal{D}}^{\ell}(f) = \mathbf{E}_{(\boldsymbol{x},y)\sim\mathcal{D}}\ell(f, \boldsymbol{x}, y)$ and the empirical risk $\hat{\mathcal{L}}_{X,Y}^{\ell}(f) = (1/n)\sum_i \ell(f, \boldsymbol{x}_i, y_i)$. We consider two probability measures, the prior $\pi \in \mathcal{M}(\mathcal{F})$ and the posterior $\hat{\rho} \in \mathcal{M}(\mathcal{F})$. Here, $\mathcal{M}(\mathcal{F})$ denotes the set of all probability measures on $\mathcal{F}$. We encounter cases where we make predictions using the posterior predictive distribution $\mathbf{E}_{f\sim\hat{\rho}}[p(y|\boldsymbol{x}, f)]$. We will use two loss functions, the non-differentiable zero-one loss $\ell_{01}(f, \boldsymbol{x}, y) = \mathbb{I}(\arg\max_j f(\boldsymbol{x})_j \neq y)$, and the negative log-likelihood, which is a commonly used differentiable surrogate $\ell_{\text{nll}}(f, \boldsymbol{x}, y) = -\log(p(y|\boldsymbol{x}, f))$, where we assume that the outputs of $f$ are normalized to form a probability distribution. Given the above, the Evidence Lower Bound (ELBO) has the following form

$$-\mathbf{E}_{f\sim\hat{\rho}}\hat{\mathcal{L}}_{X,Y}^{\ell_{\text{nll}}}(f) - \frac{1}{\lambda n}\text{KL}(\hat{\rho}\|\pi), \tag{1}$$

where $\lambda = 1$. Note that our temperature parameter $\lambda$ is the *inverse* of the one typically used in cold posterior papers. In this form $\lambda$ has a clearer interpretation as the temperature of a log-Laplace transform. Overall our setup is one of the cases discussed in Wenzel et al. [38], p3 Section 2.3. While they use MCMC to conduct their experiments, we opt for the ELBO for analytical tractability. Wenzel et al. [38] also temper by $\lambda$ both the likelihood and the prior in the MCMC inference setting. As discussed in Aitchison [2] and Wenzel et al. [38] the relevant setting for the ELBO is the one we consider (Eq. 1), where only the KL is tempered. One then typically models the posterior and prior distributions over weights using a parametric distribution (commonly a Gaussian) and optimizes the ELBO, using the reparametrization trick, to find the posterior distribution [8, 23, 31, 5, 38]. The cold posterior is the following observation:

*Even though the ELBO has the form (1) with $\lambda = 1$, practitioners have found that much larger values $\lambda \gg 1$ typically result in better test time performance, for example a lower test misclassification rate and lower test negative log-likelihood.*

## 2.1 ELBO

We assume a training sample $(X, Y) \sim \mathcal{D}^n$ as before, denote $p(\mathbf{w}|X, Y)$ the true posterior probability over predictors $f$ parameterized by $\mathbf{w}$ (typically weights for neural networks), and $\pi$ and $\hat{\rho}$ respectively the prior and variational posterior distributions as before. The ELBO results from the following calculations

$$\mathrm{KL}(\hat{\rho}(\mathbf{w})\|p(\mathbf{w}|X, Y)) = \int \hat{\rho}(\mathbf{w}) \ln \frac{\hat{\rho}(\mathbf{w})}{p(\mathbf{w}|X, Y)} d\mathbf{w} = \int \hat{\rho}(\mathbf{w}) \ln \frac{\hat{\rho}(\mathbf{w})p(Y|X)}{\pi(\mathbf{w})p(Y|X, \mathbf{w})} d\mathbf{w}$$

$$= \int \hat{\rho}(\mathbf{w}) \left[ -\ln p(Y|X, \mathbf{w}) + \ln \frac{\hat{\rho}(\mathbf{w})}{\pi(\mathbf{w})} + \ln p(Y|X) \right] d\mathbf{w}$$

$$= -n \underbrace{\left( -\mathbf{E}_{f \sim \hat{\rho}} \hat{\mathcal{L}}_{X,Y}^{\ell_{\mathrm{nll}}}(f) - \frac{1}{n} \mathrm{KL}(\hat{\rho}\|\pi) \right)}_{\mathrm{ELBO}} + \ln p(Y|X).$$

Thus, maximizing the ELBO can be seen as minimizing the KL divergence between the true posterior and the variational posterior over the weights $\mathrm{KL}(\hat{\rho}(\mathbf{w})\|p(\mathbf{w}|X, Y))$, and doesn't directly bound the test misclassification error. We could ignore this problem if consistency theorems hold and the posterior quickly contracts to a Dirac delta on the true parameters. However, operating in the regime of misspecification $f^* \notin \mathcal{F}$ and where $n$ is (comparatively) small invalidates consistency theorems such as the Blackwell–Dubins [7] theorem (for example, neural networks have multiple minima, implying misspecification). This makes it important to derive a more precise certificate of generalization through a generalization bound, which directly bounds the true risk. In the following we focus on analyzing a PAC-Bayes bound generalization bound in order to obtain insights into when the cold posterior effect occurs.

## 2.2 PAC-Bayes

For classification tasks, we are typically mainly interested in achieving low expected zero-one risk $\mathbf{E}_{f \sim \hat{\rho}} \mathcal{L}_{\mathcal{D}}^{\ell_{01}}(f)$. The ELBO objective is not directly related to this risk, however in the PAC-Bayesian literature there exist bounds specifically adapted to it. In the following we will use one of the tightest and most commonly used bounds, the "Catoni" bound, denoted $\mathcal{B}_{\mathrm{Catoni}}$.

**Theorem 1** ($\mathcal{B}_{\mathrm{Catoni}}$, [9]). *Given a distribution $\mathcal{D}$ over $\mathcal{X} \times \mathcal{Y}$, a hypothesis set $\mathcal{F}$, the 0-1 loss function $\ell_{01} : \mathcal{F} \times \mathcal{X} \times \mathcal{Y} \to [0, 1]$, a prior distribution $\pi$ over $\mathcal{F}$, a real number $\delta \in (0, 1]$, and a real number $\lambda > 0$, with probability at least $1 - \delta$ over the choice of $(X, Y) \sim \mathcal{D}^n$, we have*

$$\forall \hat{\rho} \text{ on } \mathcal{F} : \mathbf{E}_{f \sim \hat{\rho}} \mathcal{L}_{\mathcal{D}}^{\ell_{01}}(f) \leq \Phi_\lambda^{-1} \left( \mathbf{E}_{f \sim \hat{\rho}} \hat{\mathcal{L}}_{X,Y}^{\ell_{01}}(f) + \frac{1}{\lambda n} \left[ \mathrm{KL}(\hat{\rho}\|\pi) + \ln \frac{1}{\delta} \right] \right), \qquad (2)$$

*where $\Phi_\lambda^{-1}(x) = \frac{1 - e^{-\lambda x}}{1 - e^{-\lambda}}$.*

The empirical risk term is the empirical mean of the loss of the classifier over all training samples. The KL term is the complexity of the model, which in this case is measured as the KL-divergence between the posterior and prior distributions. The Moment term has been absorbed in this case in the function $\Phi_\lambda^{-1}(x) = \frac{1 - e^{-\lambda x}}{1 - e^{-\lambda}}$.

## 2.3 Safe-Bayes and other relevant work

Germain et al. [16] were the first to find connections between PAC-Bayes and Bayesian inference. However they only investigate the case where $\lambda = 1$. After identifying two sources of misspecification, Grünwald and Langford [18] proposed a solution, through an approach which they named Safe-Bayes [17, 19]. Safe-Bayes corresponds to finding a temperature parameter $\lambda$ for a generalized (tempered) posterior distribution with $\lambda$ possibly different than 1. The optimal value of $\lambda$ is found by taking a sequential view of Bayesian inference. By contrast we provide an analytical expression of the bound on true risk, given $\lambda$, and also numerically investigate the case of $\lambda > 1$. Our analysis thus provides intuition regarding which parameters (for example the curvature) might result in cold posteriors.

# 3 Experiments on classification tasks

The ELBO (1) is minimized at the probability density $\rho^\star(f)$ given by: $\rho^\star(f) := \pi(f)e^{-\lambda n \hat{\mathcal{L}}_{X,Y}^{\ell_{\mathrm{nll}}}(f)}/\mathbf{E}_{f\sim\pi}\left[e^{-\lambda n \hat{\mathcal{L}}_{X,Y}^{\ell_{\mathrm{nll}}}(f)}\right]$ [9]. We will use the Laplace approximation to the posterior in our experiments. This is equivalent to approximating $\lambda n \hat{\mathcal{L}}_{X,Y}^{\ell_{\mathrm{nll}}}(f)$ using a second order Taylor expansion around a minimum $\mathbf{w}_{\hat{\rho}}$, such that $\lambda n \hat{\mathcal{L}}_{X,Y}^{\ell_{\mathrm{nll}}}(f_{\mathbf{w}}) \approx \lambda n \hat{\mathcal{L}}_{X,Y}^{\ell_{\mathrm{nll}}}(f_{\mathbf{w}_{\hat{\rho}}}) + \lambda n(\mathbf{w} - \mathbf{w}_{\hat{\rho}})^\top \frac{1}{2}\nabla\nabla\hat{\mathcal{L}}_{X,Y}^{\ell_{\mathrm{nll}}}(f_{\mathbf{w}})|_{\mathbf{w}=\mathbf{w}_{\hat{\rho}}}(\mathbf{w} - \mathbf{w}_{\hat{\rho}})$. Assuming a Gaussian prior $\pi = \mathcal{N}(0, \sigma_\pi^2 \mathbf{I})$, the Laplace approximation to the posterior $\hat{\rho}$ is again a Gaussian

$$\hat{\rho} = \mathcal{N}\left(\mathbf{w}_{\hat{\rho}}, \left(\lambda\mathbf{H} + \frac{1}{\sigma_\pi^2}\mathbf{I}\right)^{-1}\right)$$

where $\mathbf{H}$ is the network Hessian $\mathbf{H} = n\nabla\nabla\hat{\mathcal{L}}_{X,Y}^{\ell_{\mathrm{nll}}}(f_{\mathbf{w}})|_{\mathbf{w}=\mathbf{w}_{\hat{\rho}}}$. This Hessian is generally infeasible to compute in practice for modern deep neural networks, such that many approaches employ the generalized Gauss–Newton (GGN) approximation $\mathbf{H}^{\mathrm{GGN}} := \sum_{i=1}^{n} \mathcal{J}_{\mathbf{w}}(\boldsymbol{x}_i)^\top \boldsymbol{\Lambda}(\boldsymbol{y}_i; f_i)\mathcal{J}_{\mathbf{w}}(\boldsymbol{x}_i)$, where $\mathcal{J}_{\mathbf{w}}(\boldsymbol{x})$ is the network per-sample Jacobian $[\mathcal{J}_{\mathbf{w}}(\boldsymbol{x})]_c = \nabla_{\mathbf{w}}f_c(\boldsymbol{x}; \mathbf{w}_{\hat{\rho}})$, and $\boldsymbol{\Lambda}(\boldsymbol{y}; f) = -\nabla_{ff}^2 \log p(\boldsymbol{y}; f)$ is the per-input noise matrix [25]. We will use two simplified versions of the GGN

- An isotropic approximation with variance $\sigma_{\hat{\rho}}^2(\lambda)$ such that $\frac{1}{\sigma_{\hat{\rho}}^2(\lambda)} = \frac{\lambda h}{d} + \frac{1}{\sigma_\pi^2}$, where $h = \sum_{i,j,k} g(i,k)(\nabla_{\mathbf{w}}f_k(\boldsymbol{x}_i; \mathbf{w}_{\hat{\rho}})_j)^2$ is the trace of the Gauss–Newton approximation to the Hessian, with $g(i,k) = [\boldsymbol{\Lambda}(\boldsymbol{y}_i; f)]_{kk}$.
- The Kronecker-Factorized Approximate Curvature (KFAC) [29] approximation, which retains only a block diagonal part of the GGN.

When making predictions, we use the posterior predictive distribution $\mathbf{E}_{\mathbf{w}\sim\hat{\rho}}[p(y|\boldsymbol{x}, f_{\mathbf{w}})]$ of the *full neural network model*, meaning that samples from $\hat{\rho}$ are inputted to the full neural network. Since the 0-1 loss is not differentiable, the posterior estimated with the cross entropy loss will be used for classification problems.

We have tested extensively in classification tasks, scaling from simplified settings to realistic models and datasets. For the classification task we used the CIFAR-10, CIFAR-100 [24], SVHN [34] and FashionMnist [39] datasets. In all experiments, we split the dataset into two sets. These three are the typical prediction tasks sets: training set $Z_{\mathrm{train}}$, testing set $Z_{\mathrm{test}}$, and validation set $Z_{\mathrm{validation}}$. We use Monte Carlo sampling to estimate the Empirical Risk term ($f \sim \hat{\rho}$). For the isotropic Laplace approximation, and a Gaussian isotropic prior, the KL divergence has a simple analytical expression $\mathrm{KL}(\hat{\rho}||\pi) = \frac{1}{2}\left(d\frac{\sigma_{\hat{\rho}}^2(\lambda)}{\sigma_\pi^2} + \frac{1}{\sigma_\pi^2}\|\mathbf{w}_{\hat{\rho}} - \mathbf{w}_\pi\|^2 - d - d\ln\sigma_{\hat{\rho}}^2(\lambda) + d\ln\sigma_\pi^2\right)$. PAC-Bayes bounds require correct control of the prior mean as the $\ell_2$ distance between prior and posterior means in the KL term is often the dominant term in the bound. To control this distance, we follow a variation of the approach in Dziugaite et al. [14] to constructing our classifiers. We first use $Z_{\mathrm{train}}$ to find a prior mean $\mathbf{w}_\pi$. We then set the posterior mean equal to the prior mean $\mathbf{w}_{\hat{\rho}} = \mathbf{w}_\pi$ but evaluate the r.h.s of the bounds on $Z_{\mathrm{validation}}$. Note that in this way $\|\mathbf{w}_{\hat{\rho}} - \mathbf{w}_\pi\|_2^2 = 0$, while the bound is still valid since the prior is independent from the evaluation set $X, Y = Z_{\mathrm{validation}}$. For the CIFAR-10, CIFAR-100, and SVHN datasets, we use a WideResNet22 [40], with Fixup initialization [43]. For the FashionMnist dataset, we use a convolutional architecture with three convolutional layers, followed by two fully connected non-linear layers. More details on the experimental setup can be found in the Appendix.

## 3.1 Classification experiments

We find ten MAP estimates for the neural network weights of the CIFAR-10, CIFAR-100, SVHN and FMNIST datasets by training on $Z_{\mathrm{train}}$ using SGD. We then fit an Isotropic Laplace approximation to each MAP estimate using $X, Y = Z_{\mathrm{validation}}$. For different values of $\lambda$ we then estimate the Catoni bound (Theorem 1) using $Z_{\mathrm{validation}}$. We also estimate the *test* 0-1 Loss, negative log-likelihood (NLL) and the Expected Calibration Error (ECE) [33] of the posterior predictive on $Z_{\mathrm{test}}$. We use the prior variance $\sigma_\pi^2 = 0.1$, as optimizing the marginal likelihood leads to $\sigma_\pi^2 \approx 0$ which is not relevant for BNNs. We also test two standard setups of increasing difficulty. First, the standard

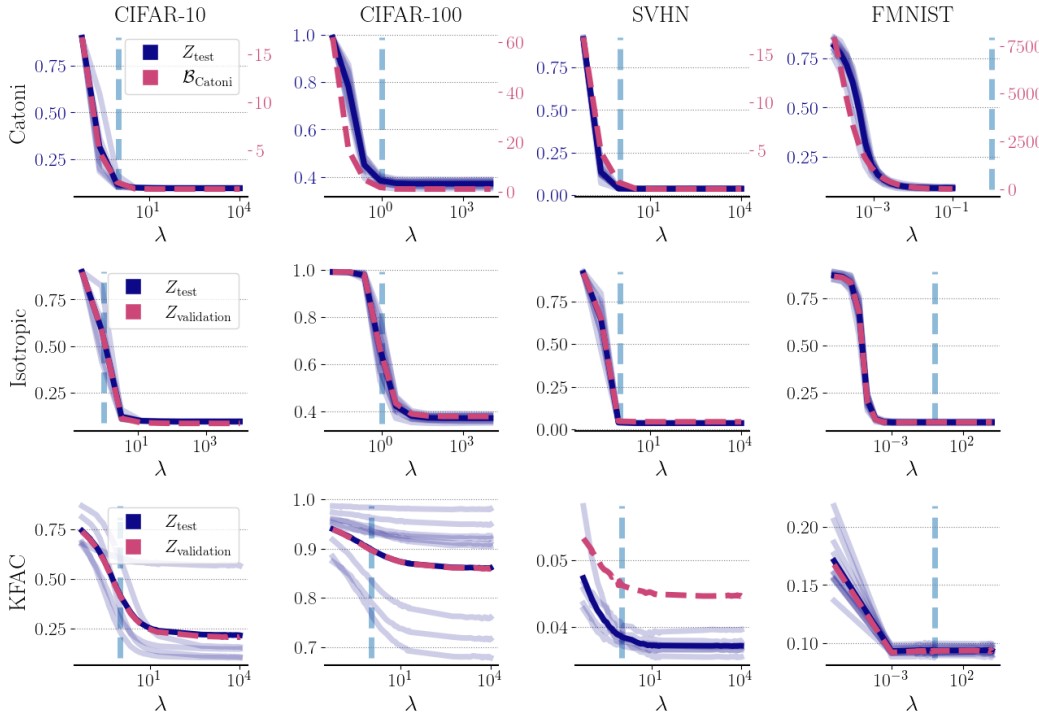

Figure 2: Test 0-1 Loss ████ mean, as well as 10 MAP trials ████, along with the generalization certificate ▬ ▬ ▬ (we denote $\lambda = 1$ by ▬ ▬ ▬): $\mathcal{B}_{\text{Catoni}}$ PAC-Bayes bound (top), standard Isotropic Laplace posterior (middle) and standard KFAC (bottom). The $\mathcal{B}_{\text{Catoni}}$ PAC-Bayes bound closely tracks the test 0-1 Loss. For the standard Isotropic and KFAC posteriors the test and validation 0-1 Loss behave similar to the Catoni case, with a rapid improvement as $\lambda \uparrow$ followed by a plateau. Coldest posteriors $\lambda \gg 1$ are always best.

"Isotropic" case where we fit the Laplace on $Z_{\text{train}}$. Second, the KFAC case where we fit the Laplace on $Z_{\text{train}}$ and also choose the prior through the marginal likelihood. In both of these last two cases, we estimate the evaluation metrics on the validation set $Z_{\text{validation}}$ as from the literature we know that any PAC-Bayes bound will be vacuous (larger than 1) as we do not control $\|\mathbf{w}_{\hat{\rho}} - \mathbf{w}_\pi\|_2^2$.

We plot the results for all datasets in Figure 2. The Catoni bound correlates tightly with test 0-1 Loss for all datasets and we plot this correlation in Figure 1(b). Again, in terms of test 0-1 Loss, the MAP estimate (obtained where $\lambda \gg 1$ and the posterior is "coldest") is optimal. This bevaviour is replicated both in the "Isotropic" and "KFAC" cases. In the Laplace approximation literature for deep neural networks, there are various similar results hidden in plain sight and to the best of our knowledge *never directly addressed* [4, 10, 37].

The crucial point here is the choice of the *evaluation metric*. We plot in the Appendix the Isotropic and KFAC cases for the NLL. We find that all three cases of temperatures (cold posterior, warm posterior, as well as posterior with $\lambda = 1$) can be optimal, for varying datasets. We discuss the ECE results in the Appendix.

## 4 Effect of temperature parameter $\lambda$ on PAC-Bayes bound

In light of our empirical results, it would be interesting to derive an analytical form that elucidates the important variables that affect the bound. However, PAC-Bayes objectives are difficult to analyze theoretically for the non-convex case. Thus in the following we make a number of simplifying assumptions. The Laplace approximation with the Generalized Gauss-Newton approximation to the Hessian corresponds to a linearization of the neural network around the MAP estimate $\mathbf{w}_{\hat{\rho}} \in \mathbb{R}^d$ [20]

$$f_{\text{lin}}(\boldsymbol{x}; \mathbf{w}) = f(\boldsymbol{x}; \mathbf{w}_{\hat{\rho}}) + \nabla_{\mathbf{w}} f(\boldsymbol{x}; \mathbf{w}_{\hat{\rho}})^\top (\mathbf{w} - \mathbf{w}_{\hat{\rho}}). \tag{3}$$

When analyzing minima of the loss landscape linearization is reasonable even without assuming infinite width Zancato et al. [41], Maddox et al. [28]. For appropriate modelling choices, we aim at deriving a bound for this linearized model.

We adopt the linear form (3) together with the Gaussian likelihood with $\sigma = 1$, yielding $\ell_{\text{nll}}(\mathbf{w}, \boldsymbol{x}, y) = \frac{1}{2}\ln(2\pi) + \frac{1}{2}(y - f(\boldsymbol{x}; \mathbf{w}_{\hat{\rho}}) - \nabla_{\mathbf{w}} f(\boldsymbol{x}; \mathbf{w}_{\hat{\rho}})^{\top}(\mathbf{w} - \mathbf{w}_{\hat{\rho}}))^2$. We also make the following modeling choices

- Prior over weights: $\mathbf{w} \sim \mathcal{N}(\mathbf{w}_{\pi}, \sigma_{\pi}^2 \mathbf{I})$.

- Gradients as Gaussian mixture: $\nabla_{\mathbf{w}} f(\boldsymbol{x}; \mathbf{w}_{\hat{\rho}}) \sim \sum_{i=1}^{k} \phi_i \mathcal{N}(\boldsymbol{\mu}_i, \sigma_{\boldsymbol{x}i}^2 \mathbf{I})$; note that this assumption should be plausible for *trained* neural networks, in that previous works have shown that per sample gradients with respect to the weights, at $\mathbf{w}_{\hat{\rho}}$, are clusterable [41]. We consider that a Gaussian Mixture model for these clusters is reasonable.

- Labeling function: $y = f(\boldsymbol{x}; \mathbf{w}_{\hat{\rho}}) + \nabla_{\mathbf{w}} f(\boldsymbol{x}; \mathbf{w}_{\hat{\rho}})^{\top}(\mathbf{w}_* - \mathbf{w}_{\hat{\rho}}) + \epsilon$, where $\epsilon \sim \mathcal{N}(0, \sigma_{\epsilon}^2)$.

Thus $y|\boldsymbol{x} \sim \mathcal{N}(f(\boldsymbol{x}; \mathbf{w}_{\hat{\rho}}) + \nabla_{\mathbf{w}} f(\boldsymbol{x}; \mathbf{w}_{\hat{\rho}})^{\top}(\mathbf{w}_* - \mathbf{w}_{\hat{\rho}}), \sigma_{\epsilon}^2)$. The assumption that $\mathbf{w}_*$ is close to $\mathbf{w}_{\hat{\rho}}$ is quite strong, and we furthermore argued in the previous sections that no single $\mathbf{w}$ is truly "correct". However we note that for fine-tuning tasks linearized neural networks work remarkably well [28, 11]. It is therefore at least somewhat reasonable to assume the above oracle labelling function, in that for deep learning architectures good $\mathbf{w}$ that fit many datasets can be found close to $\mathbf{w}_{\hat{\rho}}$ in practical settings. We also assume that we have a deterministic estimate of the posterior weights $\mathbf{w}_{\hat{\rho}}$ *which we keep fixed*, and we model the posterior as $\hat{\rho} = \mathcal{N}(\mathbf{w}_{\hat{\rho}}, \sigma_{\hat{\rho}}^2(\lambda)\mathbf{I})$, similarly to our experimental section. Therefore estimating the posterior corresponds to estimating the variance $\sigma_{\hat{\rho}}^2(\lambda)$.

**Proposition 1** ($\mathcal{B}_{\text{approximate}}$)**.** *With the above modeling choices, and given a distribution $\mathcal{D}$ over $\mathcal{X} \times \mathcal{Y}$, real numbers $\delta \in (0,1]$ and $\lambda \in (0, \frac{1}{c})$ with $c = 2n\sigma_{\boldsymbol{x}}^2 \sigma_{\pi}^2$, with probability at least $1 - \delta$ over the choice $(X, Y) \sim \mathcal{D}^n$, we have*

$$\mathbf{E}_{\mathbf{w} \sim \hat{\rho}} \mathcal{L}_{\mathcal{D}}^{\ell_{\text{nll}}}(\mathbf{w})$$

$$\leq \underbrace{\frac{\|\boldsymbol{y} - f(\mathbf{X}; \mathbf{w}_{\hat{\rho}})\|_2^2}{2n} + \left(\frac{\lambda h}{d} + \frac{1}{\sigma_{\pi}^2}\right)^{-1} \frac{h}{2n} + \frac{1}{2}\ln(2\pi)}_{\text{Empirical Risk}} + \underbrace{\frac{\sigma_{\boldsymbol{x}}^2(\sigma_{\pi}^2 d + \|\mathbf{w}_*\|_2^2)}{1 - 2\lambda n \sigma_{\boldsymbol{x}}^2 \sigma_{\pi}^2} + \sigma_{\epsilon}^2}_{\text{Moment}} +$$

$$\underbrace{\frac{1}{\lambda n}\left[\frac{1}{2}\left(\frac{d}{\sigma_{\pi}^2} \frac{1}{\frac{\lambda h}{d} + \frac{1}{\sigma_{\pi}^2}} + \frac{1}{\sigma_{\pi}^2}\|\mathbf{w}_{\hat{\rho}} - \mathbf{w}_{\pi}\|_2^2 - d - d\ln\frac{1}{\frac{\lambda h}{d} + \frac{1}{\sigma_{\pi}^2}} + d\ln\sigma_{\pi}^2\right) + \ln\frac{1}{\delta}\right]}_{\text{KL}}$$

*where $h = \sum_i \sum_j (\nabla_{\mathbf{w}} f(\boldsymbol{x}_i; \mathbf{w}_{\hat{\rho}})_j)^2$ is the curvature parameter, and $\sigma_{\boldsymbol{x}}^2 = \sum_{j=1}^{k} \phi_j \sigma_{\boldsymbol{x}j}^2$ is the posterior gradient variance.*

We now make a number of observations regarding Proposition 1. Here, $h$ is the trace of the Hessian under the Gauss–Newton approximation (without a scaling factor $n$). Under the PAC-Bayesian modeling of the risk, cold posteriors are the result of a complex interaction between various parameters resulting from 1) our *model* such as the prior variance $\sigma_{\pi}^2$ and prior mean $\mathbf{w}_{\pi}$ 2) our *data* $\sigma_{\boldsymbol{x}}^2$ and $\mathbf{w}_*$ (the curvature of the minimum $h$ and the MAP estimate $\mathbf{w}_{\hat{\rho}}$ depend on the deep neural network architecture, the optimization procedure and the data). A number of works have tried to identify the causes of the cold posterior effect [35, 15], with often contradictory results, typically identifying sufficient but necessary conditions. Given the complex interactions in Proposition 1, our result might shed light on why pinpointing the exact cause is difficult in practice.

## 5 Discussion

A number of interesting questions are raised by our results. How can we link our results to the MCMC setting? Which metrics are relevant for the cold-posterior effect? For which metrics and for which approximations to the curvature is the Laplace approximation relevant for modern deep learning? We intend to answer these in future work.

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

# A Proofs main results

## A.1 Proof of Proposition 1

Recall that we model our predictor as $f_{\text{lin}}(\boldsymbol{x}; \mathbf{w}) = f(\boldsymbol{x}; \mathbf{w}_{\hat{\rho}}) - \nabla_{\mathbf{w}} f(\boldsymbol{x}; \mathbf{w}_{\hat{\rho}})^{\top}(\mathbf{w} - \mathbf{w}_{\hat{\rho}})$. Then for the choice of a Gaussian likelihood, given a training signal $\boldsymbol{x}$, a training label $y$ and weights $\mathbf{w}$, the negative log-likelihood loss takes the form $\ell_{\text{nll}}(\mathbf{w}, \boldsymbol{x}, y) = \frac{1}{2}\ln(2\pi) + \frac{1}{2}(y - f(\boldsymbol{x}; \mathbf{w}_{\hat{\rho}}) - \nabla_{\mathbf{w}} f(\boldsymbol{x}; \mathbf{w}_{\hat{\rho}})^{\top}(\mathbf{w} - \mathbf{w}_{\hat{\rho}}))^2$. We also define $\hat{\mathcal{L}}_{X,Y}^{\ell}(f) = (1/n)\sum_i \ell(f, \boldsymbol{x}_i, y_i)$. Our derivations closely follow the approach of Germain et al. [16] p.11, section A.4.

Given the above definitions and modelling choices we develop the empirical risk term

$$
\begin{aligned}
2n\mathbf{E}_{\mathbf{w}\sim\hat{\rho}}\hat{\mathcal{L}}_{X,Y}^{\ell_{\text{nll}}}(\mathbf{w}) - n\ln(2\pi) &= \mathbf{E}_{\mathbf{w}\sim\hat{\rho}}\sum_{i=1}^{n}(y_i - f(\boldsymbol{x}_i; \mathbf{w}_{\hat{\rho}}) - \nabla_{\mathbf{w}} f(\boldsymbol{x}_i; \mathbf{w}_{\hat{\rho}})^{\top}(\mathbf{w} - \mathbf{w}_{\hat{\rho}}))^2 \\
&= \mathbf{E}_{\mathbf{w}\sim\hat{\rho}}\|\boldsymbol{y} - f(\mathbf{X}; \mathbf{w}_{\hat{\rho}}) - \nabla_{\mathbf{w}} f(\mathbf{X}; \mathbf{w}_{\hat{\rho}})^{\top}(\mathbf{w} - \mathbf{w}_{\hat{\rho}})\|_2^2 \\
&= \mathbf{E}_{\mathbf{w}\sim\hat{\rho}}[\|\boldsymbol{y} - f(\mathbf{X}; \mathbf{w}_{\hat{\rho}})\|_2^2 - 2(\boldsymbol{y} - f(\mathbf{X}; \mathbf{w}_{\hat{\rho}}))^{\top}\nabla_{\mathbf{w}} f(\mathbf{X}; \mathbf{w}_{\hat{\rho}})^{\top}(\mathbf{w} - \mathbf{w}_{\hat{\rho}}) \\
&\quad + (\mathbf{w} - \mathbf{w}_{\hat{\rho}})^{\top}\nabla_{\mathbf{w}} f(\mathbf{X}; \mathbf{w}_{\hat{\rho}})\nabla_{\mathbf{w}} f(\mathbf{X}; \mathbf{w}_{\hat{\rho}})^{\top}(\mathbf{w} - \mathbf{w}_{\hat{\rho}})] \\
&= \mathbf{E}_{\mathbf{w}\sim\hat{\rho}}[\|\boldsymbol{y} - f(\mathbf{X}; \mathbf{w}_{\hat{\rho}})\|_2^2 - 2(\boldsymbol{y} - f(\mathbf{X}; \mathbf{w}_{\hat{\rho}}))^{\top}\nabla_{\mathbf{w}} f(\mathbf{X}; \mathbf{w}_{\hat{\rho}})^{\top}(\mathbf{w} - \mathbf{w}_{\hat{\rho}}) \\
&\quad + (\mathbf{w} - \mathbf{w}_{\hat{\rho}})^{\top}\left[\textstyle\sum_i \nabla_{\mathbf{w}} f(\boldsymbol{x}_i; \mathbf{w}_{\hat{\rho}})\nabla_{\mathbf{w}} f(\boldsymbol{x}_i; \mathbf{w}_{\hat{\rho}})^{\top}\right](\mathbf{w} - \mathbf{w}_{\hat{\rho}})] \\
&= \mathbf{E}_{\mathbf{w}\sim\hat{\rho}}[\|\boldsymbol{y} - f(\mathbf{X}; \mathbf{w}_{\hat{\rho}})\|_2^2] - 2(\boldsymbol{y} - f(\mathbf{X}; \mathbf{w}_{\hat{\rho}}))^{\top}\nabla_{\mathbf{w}} f(\mathbf{X}; \mathbf{w}_{\hat{\rho}})^{\top}{\color{red}\mathbf{E}_{\mathbf{w}\sim\hat{\rho}}[\mathbf{w} - \mathbf{w}_{\hat{\rho}}]} \\
&\quad + \mathbf{E}_{\mathbf{w}\sim\hat{\rho}}\left[(\mathbf{w} - \mathbf{w}_{\hat{\rho}})^{\top}\left[\textstyle\sum_i \nabla_{\mathbf{w}} f(\boldsymbol{x}_i; \mathbf{w}_{\hat{\rho}})\nabla_{\mathbf{w}} f(\boldsymbol{x}_i; \mathbf{w}_{\hat{\rho}})^{\top}\right](\mathbf{w} - \mathbf{w}_{\hat{\rho}})\right] \\
&= \|\boldsymbol{y} - f(\mathbf{X}; \mathbf{w}_{\hat{\rho}})\|_2^2 + \sigma_{\hat{\rho}}^2\left[\textstyle\sum_i \sum_j (\nabla_{\mathbf{w}} f(\boldsymbol{x}_i; \mathbf{w}_{\hat{\rho}})_j)^2\right] \\
&= \|\boldsymbol{y} - f(\mathbf{X}; \mathbf{w}_{\hat{\rho}})\|_2^2 + \sigma_{\hat{\rho}}^2 h.
\end{aligned}
$$

In the penultimate line, we have used the fact that a real number is the trace of itself as well as the cyclic property of the trace. The second summation ($\sum_j$ over the parameters of the model) results from the fact that $\hat{\rho} = \mathcal{N}(\mathbf{w}_{\hat{\rho}}, \sigma_{\hat{\rho}}^2\mathbf{I})$ is isotropic with a common scaling factor $\sigma_{\hat{\rho}}^2$. The term in blue is exactly the Gauss–Newton approximation to the Hessian of the full neural network, for the squared loss function [25, 20], and in the last line we set $h = \left[\sum_i \sum_j (\nabla_{\mathbf{w}} f(\boldsymbol{x}_i; \mathbf{w}_{\hat{\rho}})_j)^2\right]$. Since $h$ is a sum of positive numbers, taking into account that the blue term is the Gauss–Newton approximation to the Hessian and if we assume that the Gauss–Newton approximation is diagonal, then $h$ is a measure of the curvature at minimum $\mathbf{w}_{\hat{\rho}}$ of the loss landscape. We finally get

$$
\mathbf{E}_{\mathbf{w}\sim\hat{\rho}}\hat{\mathcal{L}}_{X,Y}^{\ell_{\text{nll}}}(\mathbf{w}) = \frac{\|\boldsymbol{y} - f(\mathbf{X}; \mathbf{w}_{\hat{\rho}})\|_2^2}{2n} + \frac{\sigma_{\hat{\rho}}^2 h}{2n} + \frac{1}{2}\ln(2\pi).
$$

We continue with the KL term which is known to have the following analytical expression for Gaussian prior and posterior distributions

$$
\text{KL}(\mathcal{N}(\mathbf{w}_{\hat{\rho}}, \sigma_{\hat{\rho}}^2\mathbf{I})\|\mathcal{N}(\mathbf{w}_{\pi}, \sigma_{\pi}^2\mathbf{I})) = \frac{1}{2}\left(d\frac{\sigma_{\hat{\rho}}^2}{\sigma_{\pi}^2} + \frac{1}{\sigma_{\pi}^2}\|\mathbf{w}_{\hat{\rho}} - \mathbf{w}_{\pi}\|^2 - d - d\ln\frac{\sigma_{\hat{\rho}}^2}{\sigma_{\pi}^2}\right).
$$

We finally develop the moment term. Using an intermediate variable $\lambda_n = \frac{\lambda n}{2}$ to simplify the calculations, we get

$$\Psi_{\ell,\pi,\mathcal{D}}(\lambda, n) = \ln \mathbf{E}_{f\sim\pi}\mathbf{E}_{(X',Y')\sim\mathcal{D}^n} \exp\left[\lambda n\left(\mathcal{L}_{\mathcal{D}}^{\ell_{\mathrm{nll}}}(f) - \hat{\mathcal{L}}_{X',Y'}^{\ell_{\mathrm{nll}}}(f)\right)\right]$$

$$= \ln \mathbf{E}_{f\sim\pi}\mathbf{E}_{(X',Y')\sim\mathcal{D}^n} \exp\left[\lambda_n\left(\mathbf{E}_{(\boldsymbol{x},y)}\left[\ln(2\pi) + (y - f_{\mathrm{lin}}(\boldsymbol{x};\mathbf{w})^2\right]\right.\right.$$
$$\left.\left. - \ln(2\pi) - (1/n)\sum_i(y_i - f_{\mathrm{lin}}(\boldsymbol{x}_i;\mathbf{w})^2)\right)\right]$$

$$= \ln \mathbf{E}_{f\sim\pi}\mathbf{E}_{(X',Y')\sim\mathcal{D}^n} \exp\left[\lambda_n\left(\mathbf{E}_{(\boldsymbol{x},y)}\left[(y - f_{\mathrm{lin}}(\boldsymbol{x};\mathbf{w})^2\right] - (1/n)\sum_i(y_i - f_{\mathrm{lin}}(\boldsymbol{x}_i;\mathbf{w})^2)\right)\right]$$

$$\leq \ln \mathbf{E}_{\mathbf{w}\sim\pi} \exp\left[\lambda_n\mathbf{E}_{(\boldsymbol{x},y)}\left(y - f_{\mathrm{lin}}(\boldsymbol{x};\mathbf{w})\right)^2\right]$$

$$= \ln \mathbf{E}_{\mathbf{w}\sim\pi} \exp[\lambda_n\mathbf{E}_{(\boldsymbol{x},y)}(f(\boldsymbol{x};\mathbf{w}_{\hat{\rho}}) + \nabla_{\mathbf{w}}f(\boldsymbol{x};\mathbf{w}_{\hat{\rho}})^\top(\mathbf{w}_* - \mathbf{w}_{\hat{\rho}}) + \epsilon$$
$$- (f(\boldsymbol{x};\mathbf{w}_{\hat{\rho}}) + \nabla_{\mathbf{w}}f(\boldsymbol{x};\mathbf{w}_{\hat{\rho}})^\top(\mathbf{w} - \mathbf{w}_{\hat{\rho}})))^2]$$

$$= \ln \mathbf{E}_{\mathbf{w}\sim\pi} \exp[\lambda_n\mathbf{E}_{(\boldsymbol{x},y)}(\nabla_{\mathbf{w}}f(\boldsymbol{x};\mathbf{w}_{\hat{\rho}})^\top(\mathbf{w}_* - \mathbf{w}) + \epsilon)^2]$$

$$= \ln \mathbf{E}_{\mathbf{w}\sim\pi} \exp[\lambda_n\mathbf{E}_{\boldsymbol{x}}[(\nabla_{\mathbf{w}}f(\boldsymbol{x};\mathbf{w}_{\hat{\rho}})^\top(\mathbf{w}_* - \mathbf{w}))^2] + \lambda_n\sigma_\epsilon^2].$$

Inequality in line 4 is because the exponential function is less than 1 on the negative half line. In the fifth line we use our modelling choice $y = f(\boldsymbol{x};\mathbf{w}_{\hat{\rho}}) + \nabla_{\mathbf{w}}f(\boldsymbol{x};\mathbf{w}_{\hat{\rho}})^\top(\mathbf{w}_* - \mathbf{w}_{\hat{\rho}}) + \epsilon$, where $\epsilon \sim \mathcal{N}(0, \sigma_\epsilon^2)$. To obtain the final line we note that the gradient of the *neural network output* with respect to $\mathbf{w}$, that is $\nabla_{\mathbf{w}}f(\boldsymbol{x};\mathbf{w}_{\hat{\rho}})$, does *not* depend on the label $y$. We get the last line by applying the square and taking the expectation, given that the noise $\epsilon$ is centered.

We now take into account the Gaussian mixture modelling for the gradients per data sample, $\nabla_{\mathbf{w}}f(\boldsymbol{x};\mathbf{w}_{\hat{\rho}}) \sim \sum_{j=1}^k \phi_j\mathcal{N}(\boldsymbol{\mu}_j, \sigma_{\boldsymbol{x}j}^2\mathbf{I})$. We get

$$\mathbf{E}_{\boldsymbol{x}}[(\nabla_{\mathbf{w}}f(\boldsymbol{x};\mathbf{w}_{\hat{\rho}})^\top(\mathbf{w}_* - \mathbf{w}))^2] = \mathbf{E}_{\boldsymbol{x}}[(\sum_i \nabla_{\mathbf{w}}f(\boldsymbol{x};\mathbf{w}_{\hat{\rho}})_i(\mathbf{w}_* - \mathbf{w})_i)^2]$$

$$= \mathbf{E}_{\boldsymbol{x}}\left[(\sum_i \nabla_{\mathbf{w}}f(\boldsymbol{x};\mathbf{w}_{\hat{\rho}})_i^2(\mathbf{w}_* - \mathbf{w})_i^2 + {\color{red}2\sum_{i,j}\nabla_{\mathbf{w}}f(\boldsymbol{x};\mathbf{w}_{\hat{\rho}})_i\nabla_{\mathbf{w}}f(\boldsymbol{x};\mathbf{w}_{\hat{\rho}})_j(\mathbf{w}_* - \mathbf{w})_i(\mathbf{w}_* - \mathbf{w})_j}\right]$$

$$= \sum_i \mathbf{E}_{\boldsymbol{x}}[\nabla_{\mathbf{w}}f(\boldsymbol{x};\mathbf{w}_{\hat{\rho}})_i^2](\mathbf{w}_* - \mathbf{w})_i^2 = \sum_i \sum_{j=1}^k(\phi_j\sigma_{\boldsymbol{x}j}^2)(\mathbf{w}_* - \mathbf{w})_i^2 = \sigma_{\boldsymbol{x}}^2\|\mathbf{w}_* - \mathbf{w}\|_2^2.$$

The red term cancels out because we assumed that each weight is independent from the others. Next we use the Gaussian mixture modelling to get $\mathbf{E}_{\boldsymbol{x}}[\nabla_{\mathbf{w}}f(\boldsymbol{x};\mathbf{w}_{\hat{\rho}})_i^2] = \sum_{j=1}^k(\phi_j\sigma_{\boldsymbol{x}j}^2)$, and we finally set $\sigma_{\boldsymbol{x}}^2 = \sum_{j=1}^k(\phi_j\sigma_{\boldsymbol{x}j}^2)$, as each component of the mixture is isotropic, thus the second moment of all weights is the same. By completing the square above, one obtains the Gaussian expectation of this squared norm and forms the moment term as follows

$$\Psi_{\ell,\pi,\mathcal{D}}(\lambda, n) = \ln \mathbf{E}_{\mathbf{w}\sim\pi} \exp\left[\lambda_n\sigma_{\boldsymbol{x}}^2\|\mathbf{w}_* - \mathbf{w}\|_2^2 + \lambda_n\sigma_\epsilon^2\right]$$

$$= \ln\left(\frac{1}{(1 - 2\lambda_n\sigma_{\boldsymbol{x}}^2\sigma_\pi^2)^{\frac{d}{2}}}\exp\left[\frac{\lambda_n\sigma_{\boldsymbol{x}}^2\|\mathbf{w}_* - \mathbf{w}_\pi\|_2^2}{1 - 2\lambda_n\sigma_{\boldsymbol{x}}^2\sigma_\pi^2} + \lambda_n\sigma_\epsilon^2\right]\right)$$

$$= -\frac{d}{2}\ln(1 - 2\lambda_n\sigma_{\boldsymbol{x}}^2\sigma_\pi^2) + \frac{\lambda_n\sigma_{\boldsymbol{x}}^2\|\mathbf{w}_* - \mathbf{w}_\pi\|_2^2}{1 - 2\lambda_n\sigma_{\boldsymbol{x}}^2\sigma_\pi^2} + \lambda_n\sigma_\epsilon^2$$

$$\leq \frac{\lambda_n\sigma_{\boldsymbol{x}}^2\sigma_\pi^2 d}{1 - 2\lambda_n\sigma_{\boldsymbol{x}}^2\sigma_\pi^2} + \frac{\lambda_n\sigma_{\boldsymbol{x}}^2\|\mathbf{w}_* - \mathbf{w}_\pi\|_2^2}{1 - 2\lambda_n\sigma_{\boldsymbol{x}}^2\sigma_\pi^2} + \lambda_n\sigma_\epsilon^2$$

$$= \frac{\lambda_n\sigma_{\boldsymbol{x}}^2(\sigma_\pi^2 d + \|\mathbf{w}_* - \mathbf{w}_\pi\|_2^2)}{1 - 2\lambda_n\sigma_{\boldsymbol{x}}^2\sigma_\pi^2} + \lambda_n\sigma_\epsilon^2,$$

which assumes $1 - 2\lambda_n\sigma_{\boldsymbol{x}}^2\sigma_\pi^2 > 0$. The second line above is obtained by using the moment generating function of noncentral $\chi^2$ variables, while the inequality comes from $\ln(u) < u - 1$ for $u > 1$. Setting back $\frac{\lambda n}{2}$ in place of $\lambda_n$, we get

$$\frac{1}{\lambda n}\Psi_{\ell,\pi,\mathcal{D}}(\lambda, n) \leq \frac{\sigma_{\boldsymbol{x}}^2(\sigma_\pi^2 d + \|\mathbf{w}_* - \mathbf{w}_\pi\|_2^2)}{2 - 2\lambda n 2\sigma_{\boldsymbol{x}}^2\sigma_\pi^2} + \frac{\sigma_\epsilon^2}{2}.$$

We are now ready to minimize the following objective, where the moment term is absent since it does not depend on $\sigma_{\hat{\rho}}^2$

$$\min_{\sigma_{\hat{\rho}}^2} \mathbf{E}_{\mathbf{w}\sim\hat{\rho}}\hat{\mathcal{L}}_{X,Y}^{\ell_{\mathrm{nll}}}(\mathbf{w}) + \frac{1}{\lambda n}\left[\mathrm{KL}(\mathcal{N}(\mathbf{w}_{\hat{\rho}}, \sigma_{\hat{\rho}}^2\mathbf{I})\|\mathcal{N}(\mathbf{w}_\pi, \sigma_\pi^2\mathbf{I})) + \ln\frac{1}{\delta}\right]$$

The derivative of the objective function w.r.t. $\sigma_{\hat{\rho}}^2$ simply writes

$$\frac{\partial}{\partial \sigma_{\hat{\rho}}^2} \left( \frac{\|\boldsymbol{y} - f(\mathbf{X}; \mathbf{w}_{\hat{\rho}})\|_2^2}{2n} + \frac{\sigma_{\hat{\rho}}^2 h}{2n} + \frac{1}{2}\ln(2\pi) \right.$$

$$\left. + \frac{1}{\lambda n} \left[ \frac{1}{2} \left( \frac{1}{\sigma_{\pi}^2} d\sigma_{\hat{\rho}}^2 + \frac{1}{\sigma_{\pi}^2} \|\mathbf{w}_{\hat{\rho}} - \mathbf{w}_{\pi}\|_2^2 - d - d\ln\sigma_{\hat{\rho}}^2 + d\ln\sigma_{\pi}^2 \right) + \ln\frac{1}{\delta} \right] \right)$$

$$= \frac{h}{2n} + \frac{1}{2\lambda n} \left( \frac{d}{\sigma_{\pi}^2} - \frac{d}{\sigma_{\hat{\rho}}^2} \right).$$

Now setting the above to zero we get the typical prior-to-posterior update for a Gaussian precision term

$$\frac{1}{\sigma_{\hat{\rho}}^2} = \frac{\lambda h}{d} + \frac{1}{\sigma_{\pi}^2}.$$

The proposition is proven by replacing the terms in the bound from Theorem 1 with the results derived above.

## B  Experiments

### B.1  Experimental setup

We run our experiments on GPUs of the type NVIDIA GeForce RTX2080ti, on our local cluster. The total computation time was approximately 125 GPU hours. In the following list we include the libraries and datasets that we used together with their corresponding licences

- Laplace-Redux Package [10]: MIT License
- Netcal package [26]: Apache Software License
- Pytorch package [36]: Modified BSD Licence
- CIFAR-10 dataset [24]: MIT Licence
- CIFAR-100 dataset [24]: MIT Licence
- SVHN dataset [34]: -
- FashionMnist dataset [39]: MIT Licence

### B.2  Dataset splits

For the classification datasets CIFAR-10, CIFAR-100, SVHN, FMNIST we used the standard test and train splits. We use 10% of the data for the validation set.

### B.3  Models

For the classification datasets CIFAR-10, CIFAR-100 and SVHN we use the WideResNet22 [40] architecture. Because the Laplace approximation does not interact well [4] with BatchNorm [21] we instead use Fixup Initialization Zhang et al. [43]. We train our networks using the softmax activation and the cross-entropy loss. We use the SGD optimizer with learning rate $\eta = 0.1$, weight decay 5e-4, and momentum 0.9 and 300 epochs. We furthermore divide the initial learning rate by 10, at the point of 50%, 75% and 87% of the epochs. We also use dropout with 0.4 after all the Resnet blocks. We evaluate the NLL using the cross-entropy loss.

For the classification dataset FMNIST we use a Convolutional Network with 3 nonlinear convolutional layers followed by 2 non-linear fully connected layers. We use the SGD optimizer with learning rate $\eta = 0.001$, weight decay 5e-4, and momentum 0.9 and 10 epochs. We evaluate the NLL using the cross-entropy loss.

We *do not* use data augmentation in any experiment. This partially explains the problems with the CIFAR-100 dataset. In particular, in preliminary experiments (which we include further in the

Appendix) both the CIFAR-10 and the CIFAR-100 dataset improve significantly in accuracy with data augmentation (random flips and random crops) and the matrix inversion in the CIFAR-100 KFAC case is better posed and results in significantly improved accuracy 70% over the non augmented counterpart.

|  | Average MAP Test Error |
| --- | --- |
| CIFAR-10 | 10.4% |
| CIFAR-100 | 40.6% |
| SVHN | 4.2% |
| FMNIST | 8.8% |

Table 1: In this table we plot the average test 0-1 Loss of the MAP estimates of the different networks and datasets.

### B.3.1 Additional notes on bound evaluation

We try to make our bounds as tight as possible. To do this we try to control the term $\|\mathbf{w}_{\hat{\rho}} - \mathbf{w}_{\pi}\|_2^2$ which typically dominates the bound. We follow for all tasks a variation of the approach of Dziugaite et al. [14]. Specifically we use $\mathcal{Z}_{\text{train}}$ to learn a prior mean $\mathbf{w}_{\pi}$ then we set, $\mathbf{w}_{\hat{\rho}} = \mathbf{w}_{\pi}$, such that $\|\mathbf{w}_{\hat{\rho}} - \mathbf{w}_{\pi}\|_2^2 = 0$. Note that we can still evaluate a valid bound so long as we set $(X, Y)$ in Theorem 1 to be independent of the prior mean. This is the reason why we separated a part of the training set in the form of $\mathcal{Z}_{\text{validation}}$. We thus set $(X, Y) = \mathcal{Z}_{\text{validation}}$ in Theorem 1.

In our experiments we test multiple values of $\lambda$ and $\sigma_{\pi}^2$. Typically one would need to take a union bound over a grid on these parameters so as for the generalization bound to be valid [13]. However this typically costs only logarithmically to the actual bound. We ignore these calculations as our bounds are in general quite loose anyway, and these calculations would result in additional terms would make the final bound even more complex.

For the bounds to be valid, one would typically want to show concentration inequalities such that the Monte Carlo estimates of the Empirical Risk and the Moment terms concentrate close to the true expected value with high probability. We do not provide such guarantees. Note however that, at least for the Empirical Risk term, our sample size of $m = 100$ from the posterior distribution over weights is a sample size that is typically used in practice and provides good estimates. We have tried to balance sampling sufficiently to approximate the expectation on the one hand, and also not too much such that the computations become prohibitive.

### B.4 Additional classification results

### B.4.1 NLL results

We plot in Figure 3 the standard Isotropic and standard KFAC cases for the NLL. We find that all three cases of temperatures (cold posterior, warm posterior, as well as posterior with $\lambda = 1$) can be optimal, for varying datasets. Furthermore the test behaviour is dominated again by a sharp improvement as we decrease the posterior variance ($\lambda \uparrow$) followed by a plateau. An optimal $\lambda$ strictly less than $+\infty$ (when it exists) results in only a relatively modest variation of the overall trend. Thus, we believe that our bounds would be informative even in a hypothetical scenario where they would not be able to capture these optimal $\lambda < +\infty$.

### B.4.2 ECE results

We plot in Figure 4 the standard Isotropic and standard KFAC cases for the ECE. Even without data augmentation and even when we optimize the prior variance using the marginal likelihood, we find that all three cases of temperatures (cold posterior, warm posterior, as well as posterior with $\lambda = 1$) can be optimal, for varying datasets. Unfortunately we are not aware of approaches to directly bound the ECE. In Figure 4 the ECE is notable for having a significantly different behaviour from the NLL and the 0-1 Loss.

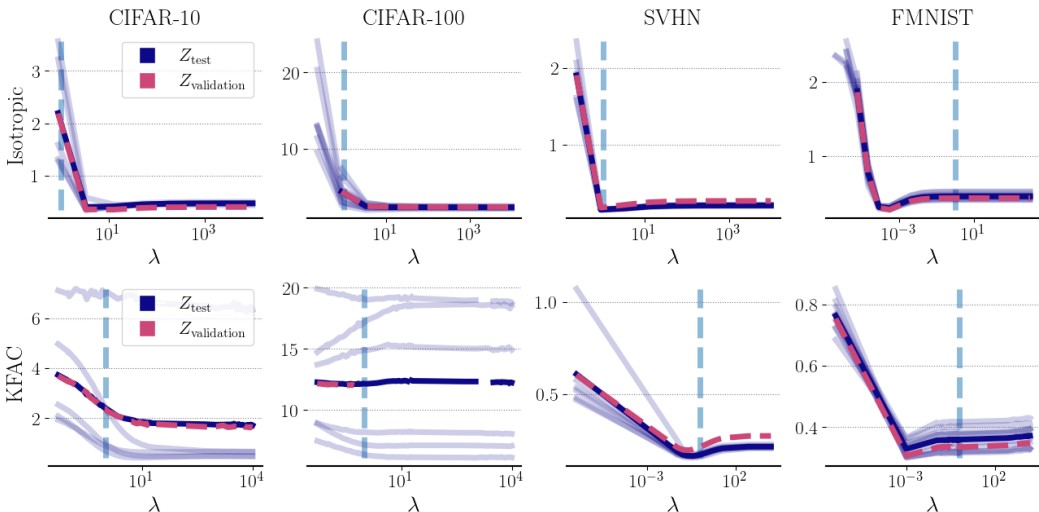

Figure 3: Test NLL ——— mean, as well as 10 MAP trials ———, along with the validation NLL ▬ ▬ ▪ (we denote $\lambda = 1$ by ▬ ▬ ▪) for the Standard Isotropic Laplace posterior (top) and standard KFAC (bottom). The test and validation NLL show warm posteriors (FMNIST and SVHN KFAC), cold posteriors (CIFAR-10) and posteriors with $\lambda = 1$ (SVHN Isotropic). The general trend remains a rapid improvement as $\lambda \uparrow$ followed by a plateau, however the coldest posteriors $\lambda \gg 1$ are not always optimal contrary to the 0-1 Loss case.

Better calibration in terms of ECE than a simple MAP estimate is one of the purported main benefits of the Bayesian paradigm. In Figure 5 we plot the Pareto front of the *test* 0-1 Loss with respect to the *test* ECE. The top row is the standard Isotropic case and the bottom row is the standard KFAC case. We see that in most cases there is a clear tradeoff between the test 0-1 Loss and the test ECE. These results might be relevant for the applicability of the Laplace approximation for improving the ECE, in that it seems that we cannot achieve a clear improvement in ECE without hurting test accuracy.

### B.4.3 Augmentation results

In Figure 6 we see that data augmentation (random flips and crops) results in better test accuracy and makes the matrix inversion in the Laplace approximation better posed such that the accuracy on CIFAR-100 is within a normal range.

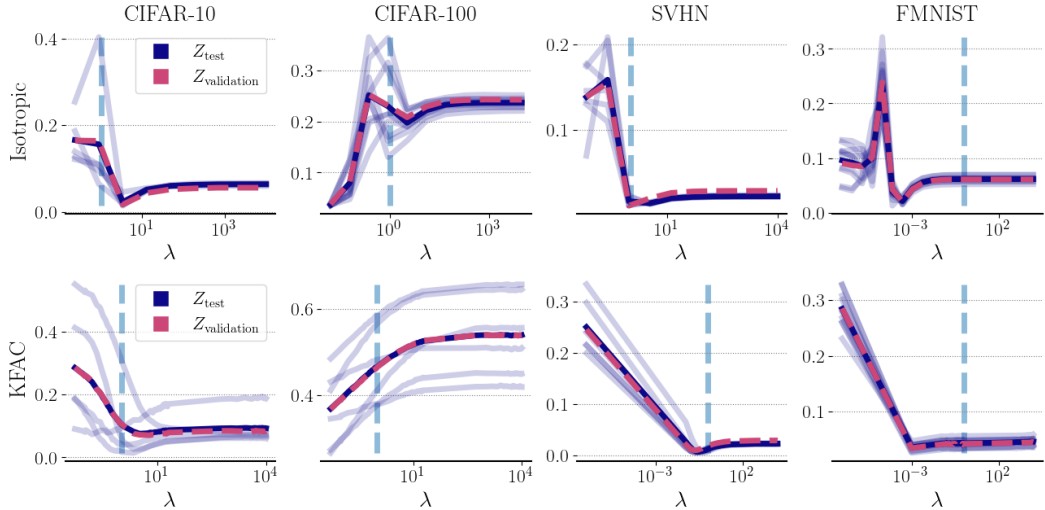

Figure 4: Test ECE ▬▬ mean, as well as 10 MAP trials ▬▬, along with the validation ECE ▬ ▬ ▬ (we denote $\lambda = 1$ by ▬ ▬ ▬) for the Standard Isotropic Laplace posterior (top) and standard KFAC (bottom). The test and validation ECE show warm posteriors (FMNIST and SVHN KFAC), cold posteriors (CIFAR-10) and posteriors with $\lambda = 1$ (SVHN Isotropic). The general trend remains a rapid improvement as $\lambda \uparrow$ followed by a plateau, however the coldest posteriors $\lambda \gg 1$ are not always optimal contrary to the 0-1 Loss case.

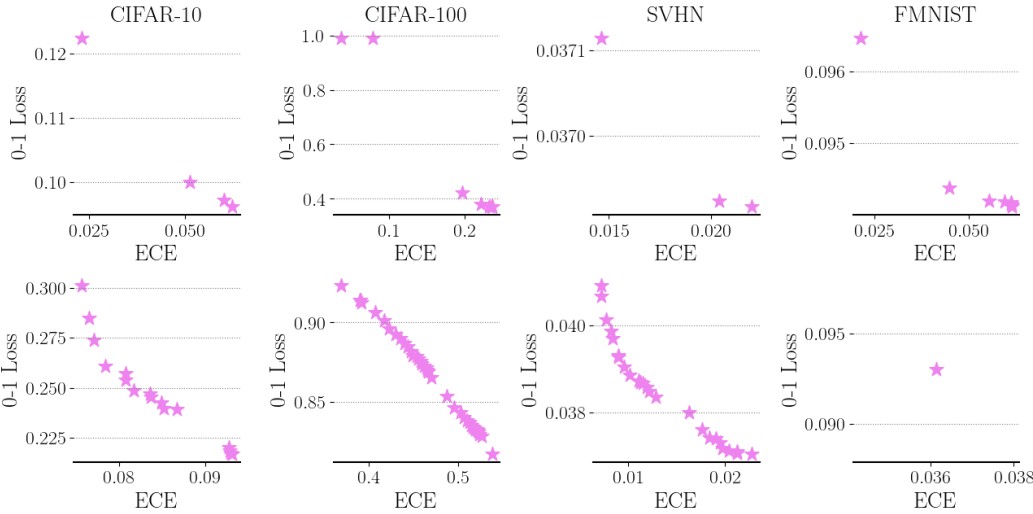

Figure 5: We plot the Pareto front of the *test* 0-1 Loss with respect to the *test* ECE. The top row is the standard Isotropic case and the bottom row is the standard KFAC case. We see that in most cases there seems to be a tradeoff between the test 0-1 Loss and the test ECE.

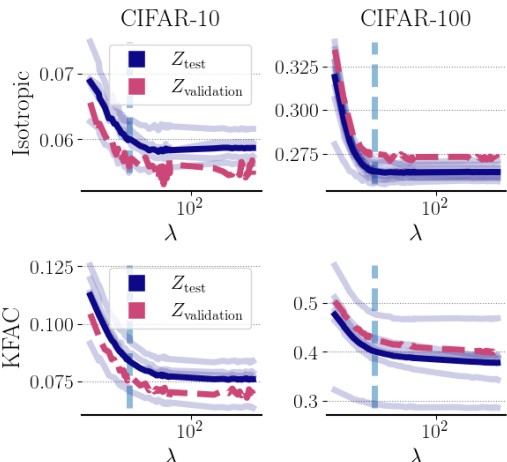

Figure 6: Test 0-1 Loss ▬▬▬ mean, as well as 10 MAP trials ▬▬▬, along with the validation 0-1 Loss ▬ ▬ ▬ (we denote $\lambda = 1$ by ▬ ▬ ▬) for the Standard Isotropic Laplace posterior (top) and standard KFAC (bottom) for CIFAR-10 and CIFAR-100 with data augmentation (random flips and crops). The performance on both improves significantly and the Laplace approximation becomes better posed.

