# OpenReview forum: "Cold Posteriors through PAC-Bayes"
_NeurIPS.cc/2022/Workshop/TSRML — TSRML2022_

### Official Review · Reviewer_XzxT · 2022-10-20
**Recomment acceptance**

**Overall Recommendation:** I recommend acceptance of this paper.
**Overall Rating:** 7

**Summary:**

This paper studies the cold posterior effect using PAC-Bayes generalization bounds. By studying the connection between the ELBO objective and the PAC-Bayes objective, the authors show that PAC-Bayes naturally contain a temperature parameter, which shows interesting features of the cold posterior effect.

**Strengths:**

(1) This paper is well-written and easy to read.
(2) The study of the cold posterior effect on PAC-Bayes is interesting.
(3) Study on the effect of temperature parameters seems promising, empirically the coldest temperature is optimal.
(4) The authors apply the Gauss-Newton Laplace approximation to derive a bound that can be potentially useful for other tasks.


**Weaknesses:**

(1) It is hard to understand Figure 1 when reading the introduction, and it seems to be referenced only in Section 3.

**Review Confidence:**

2: The reviewer is willing to defend the evaluation, but it is quite likely that the reviewer did not understand central parts of the paper

---

### Official Review · Reviewer_DoXU · 2022-10-21
**An interesting analysis of the cold posterior effect with PAC-Bayes generalization bounds**

**Overall Rating:** 7

**Summary:**

The paper analyses the cold posterior effect with a generalization-bound perspective (PAC-Bayes) in a non-asymptotic setting with fewer training samples. The paper highlights a critical issue that approximate bayesian inference methods do not provide performance guarantees on out-of-sample data which can be a critical reason behind the cold posterior effect. Thereby the authors draw an interesting connection between the variational lower bounds and PAC Bayes objective and justify the connection with theoretical and empirical analysis under certain approximations.



**Strengths:**

An interesting connection between the PAC-Bayes generalization bound and variational lower bound in the context of the cold posterior effect is quite novel and has not been explored much in literature, especially in an asymptotic setting with fewer training samples. The paper shows an explicit dependence between the generalization performance (out of sample) with the PAC-Bayes bounds for different values of the temperature which is interesting. The paper highlights that maximizing the ELBO is an approximation and doesn’t directly bind the test misclassification error which can be problematic in misspecification motivating a PAC generalization bound approach which is interesting. Experimental analysis showed that the Dirac-delta posterior on the MAP estimate outperformed in terms of accuracy compared to Isotropic and KFAC posteriors. Theoretical and empirical findings are interesting and detailed.

**Weaknesses:**

Due to the difficulty in the analysis of PAC-Bayes bounds for non-convex settings, the paper makes certain approximations including a Generalized Gauss-Newton approximation to the Hessian resulting in the linearization of the Neural network, and a bound is derived under this setting of a linearized model which is restrictive. A further discussion would have been helpful with other less restrictive assumptions.


**Overall Recommendation:**

The paper draws an interesting connection between the cold posterior effect with PAC-Bayes generalization bounds and the results are interesting and novel compared to prior research.

**Review Confidence:**

3: The reviewer is fairly confident that the evaluation is correct

---

### Decision · Program_Chairs · 2022-10-23

Accept